# Sparse Unmixing for Hyperspectral Image with Nonlocal Low-Rank Prior

**Yuhui Zheng** [1] , **Feiyang Wu** [1] , **Hiuk Jae Shim** [1] **and Le Sun** [1,2,*]

1   School of Computer and Software, Nanjing University of Information Science and Technology,
    Nanjing 210044, China; zheng_yuhui@nuist.edu.cn (Y.Z.); feiyangwu@nuist.edu.cn (F.W.);
    waitnual@nuist.edu.cn (H.J.S.)
2   Jiangsu Collaborative Innovation Center of Atmospheric Environment and Equipment Technology,
    Nanjing University of Information Science and Technology, Nanjing 210044, China
*   Correspondence: sunlecncom@nuist.edu.cn

**Abstract:** Hyperspectral unmixing is a key preprocessing technique for hyperspectral image analysis. To further improve the unmixing performance, in this paper, a nonlocal low-rank prior associated with spatial smoothness and spectral collaborative sparsity are integrated together for unmixing the hyperspectral data. The proposed method is based on a fact that hyperspectral images have self-similarity in nonlocal sense and smoothness in local sense. To explore the spatial self-similarity, nonlocal cubic patches are grouped together to compose a low-rank matrix. Then, based on the linear mixed model framework, the nuclear norm is constrained to the abundance matrix of these similar patches to enforce low-rank property. In addition, the local spatial information and spectral characteristic are also taken into account by introducing TV regularization and collaborative sparse terms, respectively. Finally, the results of the experiments on two simulated data sets and two real data sets show that the proposed algorithm produces better performance than other state-of-the-art algorithms.

**Keywords:** hyperspectral images; sparse unmixing; nonlocal self-similarity; low-rank

## 1. Introduction

Hyperspectral remote sensing generally uses an imaging spectrometer to collect spatial and spectral information. Hyperspectral images can obtain tens or even hundreds of consecutive narrow-band information (generally less than 10 nm) of each pixel in a spectral domain, and each pixel can extract a continuous spectral curve [1]. Therefore, hyperspectral images have the property of high spectral resolution, which greatly improves the ability to detect the property of the material. It has been actively discussed by researchers and widely applied in many fields, such as agricultural production, geological survey, urban planning, and environmental monitoring [2,3]. However, because of the limitations of optical instrument's performance and imperfect spectral acquisition techniques, the spatial resolution of hyperspectral images is low, which results in a pixel that may contain more than one type of ground object signature, called a mixed pixel [4,5]. Because of the existence of many mixed pixels, the accuracy of hyperspectral image processing has been greatly affected. Therefore, hyperspectral unmixing is a key preprocessing technique for hyperspectral image analysis.

The goal of hyperspectral unmixing is to extract the pure spectral signatures (called endmembers) in the scene [6] at first and then estimate the corresponding proportions (called abundances) of these endmembers [7]. There are two main models of hyperspectral unmixing: linear mixture model (LMM) [8,9] and nonlinear mixture model (NLMM) [10–12]. Different from the NLMM, the LMM has the advantages of simplicity, high efficiency, and clear physical meaning. The LMM also can describe

better the actual spectral mixing phenomenon of hyperspectral images with spatial resolution under the meter level. Therefore, the LMM is applied for hyperspectral unmixing widely and it assumes that the observed spectral signal of each mixed pixel can be approximated as a linear mixture of all pure endmembers in that pixel.

With the LMM, there are many endmember extraction algorithms based on the statistical and geometry methods that have been proposed for hyperspectral images, such as the vertex component analysis (VCA) [13], pixel purity index [14], and N-FINDR [15] algorithms. These algorithms only require a small amount of prior knowledge of the hyperspectral image. Nevertheless, they need the assumption of pure pixel's existence in the scene, which does not hold in many datasets. Thus, some researchers proposed algorithms that do not require this assumption, such as minimum volume simplex analysis (MVSA) [16] and iterative constrained endmembers (ICE) [17]. Several nonnegative matrix factorization (NMF) methods have been proposed for hyperspectral unmixing, such as minimum volume constrained nonnegative matrix factorization (MVC-NMF) [18] and robust collaborative nonnegative matrix factorization (R-CoNMF) [19]. There are also some algorithms for hyperspectral image unmixing by using the support vector machine (SVM) [20–22]. In addition, the convolutional neural network (CNN), a deep learning method, has been applied to semi-supervised learning, target detection, and other fields [23–28]. Hence, Licciardi et al. [29] proposed an auto-associative neural network for pixel unmixing based on CNN. Chen et al. [30] used CNN to extract deep characteristics of the hyperspectral image for classification. However, these algorithms perform poorly with highly mixed or noisy hyperspectral image data [31] and NMF methods may obtain virtual endmembers without physical meaning [32].

In practice, the diversity and complexity of the ground objects increase the difficulty of endmember extraction, that is to say, the endmember matrix extracted from the observed hyperspectral image is not accurate enough. To avoid the problem of unreliable abundance estimates caused by inaccurate endmember extraction during the unmixing processing, sparse unmixing, based on compressed sensing and sparse representation [33,34], has attracted more and more researchers' attention. The advantage of the sparse unmixing technology is that it does not require endmember extraction, but directly uses the spectral characteristics in a given spectral library, released by United States Geological Survey (USGS) [35], to constitutes endmember matrix, and then estimates abundance coefficients. In general, the number of spectral characteristics in the spectral library is usually far more than the number of endmembers in a given real hyperspectral scene. Therefore, the abundance coefficient vector, related with the spectral library, of each observed mixed pixel spectral signature is sparse [36]. To increase the accuracy of unmixing result, sparse unmixing introduces additional information as a priori knowledge to a hyperspectral unmixing model, which is called a regularization term.

For sparse unmixing, some researchers focus on exploring the prior information of the spectrum. Iordache et al. [31] proposed the sparse unmixing by variable splitting and augmented Lagrangian (SUnSAL) algorithm, which applies a sparse regularization term to an abundance matrix. They replaced the $L_0$ norm with $L_1$ norm and used alternating direction multiplier method (ADMM) [37] to obtain a sparse solution of the abundance matrix under the $L_1$ norm. The collaborative SUnSAL (CLSUnSAL) [38] algorithm utilizes the row sparsity characteristic of abundance coefficient for the abundance matrix, which can obtain the sparsity of abundance matrix more effectively than the $L_1$ norm. However, its performance degrades especially when an endmember is contained only in local homogeneous regions rather than the whole scene. Tang et al. [39] proposed a sparse unmixing using a priori information of the spectrum, which assumes that a part of the spectral signatures in a hyperspectral image are available before unmixing. Although unmixing results can be improved, this assumption cannot always hold. Zhang et al. [40] adopted the local collaborative sparse regression (LCSU) algorithm. They assume that endmembers are usually distributed in local homogeneous regions rather than the full image, which overcomes the limitations of CLSUnSAL.

These sparse unmixing algorithms only consider the sparseness of the spectrum, however, the spatial correlation of hyperspectral images can also improve the performance of unmixing [41].

The SUnSAL with total variation (SUnSAL-TV) [42] algorithm, proposed by Iordache et al. uses the anisotropy total variation (TV) to characterize the local spatial clustering properties of adjacent pixels, which promoted the smoothness between pixels. But this algorithm has a limitation that may cause over-smoothing in edge regions. Sun et al. proposed the $L_1$-$L_2$ SUnSAL-TV algorithm [43], which promoted the sparseness of the spectrum better than $L_1$ norm.

In recent years, the low-rank property of hyperspectral images has been taken into account to characterize the spatial correlation. The low-rank property means that in hyperspectral images, there are many regions with high similarity, in which the corresponding abundance vectors of the pixels in these regions have a high linear correlation [44], that is, the abundance matrix composed of these abundance vectors has low-rank property. In the fields of hyperspectral image restoration and denoising, some algorithms have achieved superior results by using low-rank property [45–51]. Yang et al. [52] proposed a low-rank constraint to couple sparse unmixing and denoising. They first performed unmixing and then denoising in each iteration. But, the result of unmixing depends more on the quality of the denoising task. Giampouras et al. [53] utilized sparse and low-rank properties simultaneously and proposed ADSpLRU algorithm. They imposed a low-rank constraint on the small sliding window over a hyperspectral image. Mei et al. [54] utilized superpixel segmentation and low-rank representation to unmix hyperspectral images. However, they focused on the generalized bilinear model, which is a nonlinear mixture model. Rizkinia and Okuda [55] proposed a joint local abundance sparse unmixing (J-LASU) algorithm. They used a small three-dimensional (3D) block and sliding on the 3D abundance data, which converted an abundance matrix and imposed low-rank constraint to the abundance data in this small 3D block.

Although these sparse unmixing algorithms show superiority to some extent, they have their own limitations. For example, CLSUnSAL only considers the sparseness of the spectrum, SUnSAL-TV may cause over-smoothing in the edge regions and J-LASU only considers the similarity of the image in a local region. Therefore, in this study, we propose a nonlocal low-rank prior to the sparse unmixing problem (NLLRSU). Considering that a hyperspectral image is a natural image, it has self-similarity in nonlocal regions. Thus, we propose a nonlocal low-rank regularization term to utilize nonlocal self-similarity property. First, we convert an abundance matrix to a 3D abundance cube and use a small 3D patch sliding on this abundance cube. For each small 3D patch, we find several small patches similar to this 3D patch by utilizing a block matching algorithm [56]. Then, we generate a patch group with these similar patches and use the nuclear norm to enforce the low-rank property to this patch group. In addition, we take the spectral and spatial information into account by introducing collaborative sparse and TV regularization terms, respectively. Figure 1 shows the flow chart of the proposed unmixing algorithm.

This study has four advantages.

(1) The non-local low-rank regularization can help to preserve the details of the image better than the state-of-the-art algorithms. In addition, the proposed algorithm utilizes both spectral and spatial information simultaneously to obtain better unmixing results.

(2) In order to improve unmixing performance, a large spectral library is used as an endmember matrix instead of extracting endmembers from the hyperspectral image directly.

(3) The optimization problem of the proposed algorithm with all convex terms is solved by the alternating direction multiplier method (ADMM).

(4) Extensive experiments on both simulated and real data sets validate the superiority of the proposed method in unmixing hyperspectral images.

The rest of this paper is organized as follows. In Section 2, we discuss the linear spectral unmixing and sparse unmixing problem. In Section 3, we describe the proposed NLLRSU algorithm and optimization. In Section 4, we test the proposed algorithm and other sparse unmixing algorithms with two simulated data sets and two real hyperspectral data sets. Finally, we summarize this paper in Section 5.

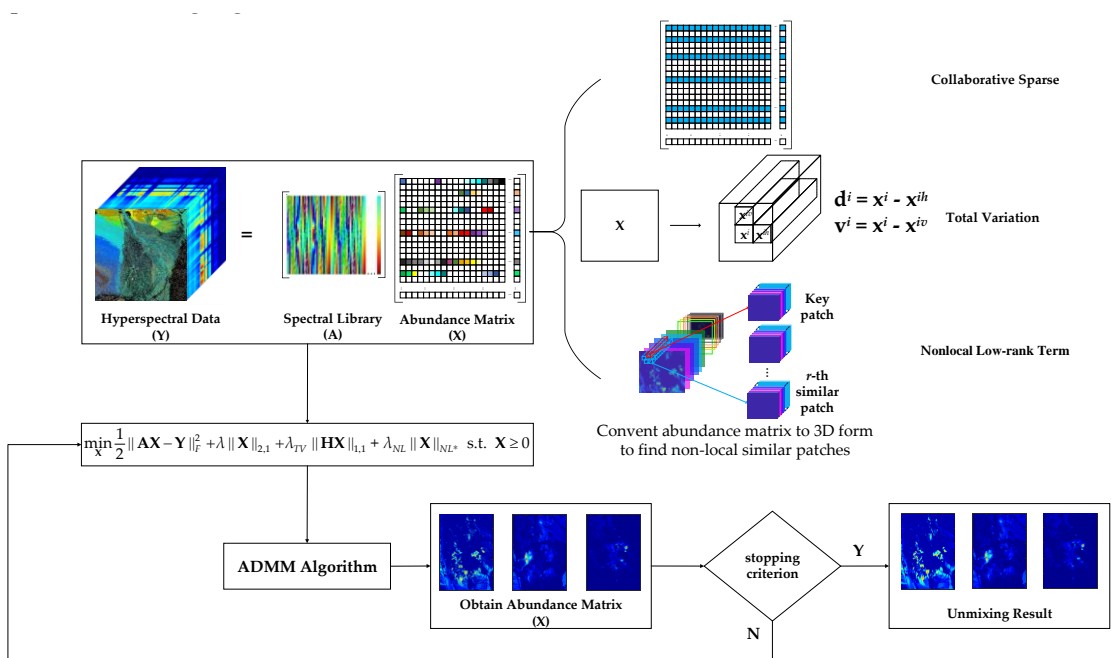

**Figure 1.** The flow chart of the proposed unmixing algorithm.

## 2. Related Works

### 2.1. Linear Spectral Unmixing

The linear mixed model (LMM) assumes that each pixel spectrum can be linearly combined by all endmembers and corresponding abundances exist in the pixel [57]. The linear model can be described as follows:

$$\mathbf{y} = \sum_{i=1}^{q} m_i \alpha_i + \mathbf{n} = \mathbf{M}\boldsymbol{\alpha} + \mathbf{n}, \tag{1}$$

where $\mathbf{y} \in \mathbb{R}^{l \times 1}$ is a column vector which represents a mixed pixel and $l$ is the number of bands, $\mathbf{M} \in \mathbb{R}^{l \times q}$ is an endmember matrix and $q$ is the number of endmembers, $\boldsymbol{\alpha} \in \mathbb{R}^{q \times 1}$ represents the abundance vector of the endmembers, $\mathbf{n} \in \mathbb{R}^{l \times 1}$ is the model error and noise that are generated during the observation process.

To ensure that the solution of the abundance has the physical meaning, the LMM introduces two constraints, named the abundance non-negativity constraint (ANC) and the abundance sum-to-one constraint (ASC) [58], which are expressed as:

$$\text{ANC} : \alpha_i \geq 0 \ (i = 1, 2, \ldots, q), \tag{2}$$

$$\text{ASC} : \sum_{i=1}^{q} \alpha_i = 1 \tag{3}$$

In a hyperspectral scene, let $\mathbf{Y} \in \mathbb{R}^{l \times s}$ be the observed hyperspectral image data and $\mathbf{X} \in \mathbb{R}^{q \times s}$ be the abundance matrix, where $s$ is the number of pixels in the hyperspectral image. Equation (1) can be rewritten as:

$$\mathbf{Y} = \mathbf{M}\mathbf{X} + \mathbf{N}, \tag{4}$$

where $\mathbf{N} \in \mathbb{R}^{l \times s}$ represents noise and model error.

## 2.2. Sparse Unmixing

The sparse unmixing utilizes a large spectral library $\mathbf{A} \in \mathbb{R}^{l \times t}$ instead of the endmember matrix, where $t$ is the number of spectral signatures of ground material in the spectral library $\mathbf{A}$. The model can be described as:

$$\mathbf{Y} = \mathbf{AX} + \mathbf{N}, \tag{5}$$

where $\mathbf{X} \in \mathbb{R}^{t \times s}$ is the abundance matrix corresponding to the spectral library $\mathbf{A}$.

The number of spectral characteristics in the spectral library $\mathbf{A}$ is usually far more than the number of endmembers in a given real hyperspectral scene. Thus, the abundance matrix $\mathbf{X}$ contains many zero values, so the abundance matrix is considered as sparse. Therefore, the sparse unmixing model can be expressed as:

$$\min_{\mathbf{X}} \frac{1}{2} \|\mathbf{AX} - \mathbf{Y}\|_F^2 + \lambda \|\mathbf{X}\|_0 \quad \text{s.t.} \ \mathbf{X} \geq 0, \tag{6}$$

where $\|\mathbf{AX} - \mathbf{Y}\|_F^2$ represents the reconstruction error of the model and $\|\mathbf{X}\|_F \equiv \sqrt{\text{trace}\{\mathbf{XX}^T\}}$ is the Frobenius norm of the $\mathbf{X}$, $\|\mathbf{X}\|_0$ is the $L_0$ norm of the abundance matrix $\mathbf{X}$, which denotes the number of non-zero elements in $\mathbf{X}$. Here, we do not introduce the ASC constraint to Equation (6) because ASC has received extensive criticism from scholars [31].

Since the $L_0$ norm is non-convex, the problem of minimization of Equation (6) is difficult to solve. Recently, one theory proved that when the spectral library matrix $\mathbf{A}$ satisfies the restricted isometry property (RIP) condition [59], the $L_0$ norm problem can be equivalently converted to the $L_1$ norm problem, which is a convex optimization problem and easier to solve. Therefore, we can replace the $L_0$ norm by the $L_1$ norm to transform the nonconvex optimization problem into a convex one. Accordingly, Equation (6) can be rewritten as:

$$\min_{\mathbf{X}} \frac{1}{2} \|\mathbf{AX} - \mathbf{Y}\|_F^2 + \lambda \|\mathbf{X}\|_1 \ \text{s.t.} \ \mathbf{X} \geq 0, \tag{7}$$

where $\|\mathbf{X}\|_1 \equiv \sum_{i=1}^{t} \sum_{j=1}^{s} |x_{ij}|$ is the $L_1$ norm which calculates the sum of the elements in the abundance matrix $\mathbf{X}$.

In practice, one hyperspectral scene often has only a few endmembers out of a large spectral library. CLSUnSAL algorithm enforces the column (pixels) of the abundance matrix $\mathbf{X}$ share the same active set of endmembers, so $\mathbf{X}$ has only a few nonzero rows (endmembers). That is, $\mathbf{X}$ has the characteristics of sparse among the rows. To utilize this prior, CLSUnSAL replaces the $L_1$ norm by $L_{2,1}$ norm which is sparser than the $L_1$ norm. The equation of CLSUnSAL algorithm is expressed as follows:

$$\min_{\mathbf{X}} \frac{1}{2} \|\mathbf{AX} - \mathbf{Y}\|_F^2 + \lambda \|\mathbf{X}\|_{2,1} \ \text{s.t.} \ \mathbf{X} \geq 0, \tag{8}$$

where $\|\mathbf{X}\|_{2,1} \equiv \sum_{i=1}^{t} \|\mathbf{x}_i\|_2 \equiv \sum_{i=1}^{t} \sqrt{\sum_{j=1}^{s} |x_{ij}|^2}$ is the $L_{2,1}$ norm of the abundance matrix $\mathbf{X}$ and $\mathbf{x}_i$ represents the $i$-th row of $\mathbf{X}$.

In addition to utilizing the properties of the spectrum, the spatial correlation of hyperspectral images is also beneficial for improving the result of the unmixing. Each pixel and its neighboring pixels usually contain similar material, so the spectral characteristic between these pixels are very similar. To promote smoothness between adjacent pixels, the TV regularization term, which describes spatial correlation, can be applied to Equation (7). Then, the model is:

$$\min_{\mathbf{X}} \frac{1}{2} \|\mathbf{AX} - \mathbf{Y}\|_F^2 + \lambda \|\mathbf{X}\|_1 + \lambda_{TV} \text{TV}(\mathbf{X}) \ \text{s.t.} \ \mathbf{X} \geq 0, \tag{9}$$

where

$$\text{TV}(\mathbf{X}) \equiv \sum_{\{i,j\} \in \varepsilon} \|\mathbf{x}^i - \mathbf{x}^j\|_1, \tag{10}$$

represent nonisotropic TV, $x^i$ and $x^j$ are *i*-th and *j*-th columns of **X**, respectively, $\varepsilon$ denotes the set that contains horizontal and vertical neighbors in a hyperspectral image [42].

Let $\mathbf{H}_h : \mathbb{R}^{t \times s} \to \mathbb{R}^{t \times s}$ and $\mathbf{H}_v : \mathbb{R}^{t \times s} \to \mathbb{R}^{t \times s}$ represent the horizontal and vertical differential linear operators of adjacent pixels in the abundance matrix **X**, respectively. $\mathbf{H}_h\mathbf{X} = [\mathbf{d}^1, \mathbf{d}^2, \dots, \mathbf{d}^s]$ calculates the difference of the **X** and horizontal neighboring pixels, where $\mathbf{d}^i = \mathbf{x}^i - \mathbf{x}^{ih}$, $\mathbf{x}^i$ is a column vector and $\mathbf{x}^{ih}$ is the horizontal neighboring column vector of $\mathbf{x}^i$. Similar to $\mathbf{H}_h\mathbf{X}$, $\mathbf{H}_v\mathbf{X} = [\mathbf{v}^1, \mathbf{v}^2, \dots, \mathbf{v}^s]$ calculates the vertical difference, where $\mathbf{v}^i = \mathbf{x}^i - \mathbf{x}^{iv}$, $\mathbf{x}^i$ is a column vector and $\mathbf{x}^{iv}$ is the vertical neighboring column vector of $\mathbf{x}^i$. By horizontal and vertical operations, we can obtain $\mathbf{HX} \equiv [\mathbf{H}_h\mathbf{X}; \mathbf{H}_v\mathbf{X}]$ [42,60]. Therefore, Equation (9) can be rewritten as:

$$\min_{\mathbf{X}} \frac{1}{2}\|\mathbf{AX} - \mathbf{Y}\|_F^2 + \lambda\|\mathbf{X}\|_1 + \lambda_{TV}\|\mathbf{HX}\|_1 \text{ s.t. } \mathbf{X} \geq 0, \tag{11}$$

However, CLSUnSAL only considers the sparsity of the abundance from a perspective in the spectral domain. SUnSAL-TV only exploits the local smoothness in the spectral domain, it may cause over smoothing in the edge regions. There is still much room for improving the unmixing performance. Therefore, we propose a nonlocal low-rank prior to the sparse unmixing (NLLRSU) algorithm to obtain more precise spatial information by exploiting the nonlocal self-similarity.

## 3. Proposed Algorithm

### 3.1. Nonlocal Self-Similarity

Treated as a natural image in each band, hyperspectral images are smooth in local regions between adjacent pixels and bands, meanwhile, they are self-similar in nonlocal regions. The adjacent pixels have smooth property. Therefore, there is a high correlation between the pixels of the local regions in the hyperspectral images, that is, the pixels are very likely to contain the same material in a local region. In addition, there are a large number of similar structural information in different regions in the hyperspectral images. These similar structures consist of smooth regions, texture regions, and edge regions. For any local region, we can find many similar regions in the image. That is to say, the information of the hyperspectral image itself is redundant and the pixels in these similar regions are also very likely to contain the same material. Qu et al. [44] indicated that this high spatial correlation of hyperspectral images means that there is also a high correlation in the abundance matrix, which is reflected by the linear correlation between the abundance vectors. Based on this prior, the abundance matrix can be reconstructed by the proposed nonlocal low-rank algorithm.

In order to take advantage of the nonlocal low-rank property, we need to search for similar structural texture regions in the abundance matrix. Therefore, we convert the abundance matrix to 3D abundance cube, which restores the position of the pixels and spectral bands in the abundance matrix to the corresponding position in the original 3D abundance cube, then we use a small 3D sliding patch on this abundance cube. For each small 3D patch, denoted by a key patch, we can find several small patches similar to this key patch in the abundance cube by utilizing a block matching algorithm [56]. Then, we stack the key patch and these similar patches to generate a patch group and use the nuclear norm to enforce the low-rank property to this patch group. To sum up, the small 3D patch slides across all dimensions of the abundance cube and the proposed nonlocal low-rank regularization term employs the nuclear norm to enforce the low-rank property of each patch group.

In addition, the results of sparse unmixing are affected by the correlation between spectral characteristics in the spectral library [61]. In general, a spectral library contains many spectral characteristics of the same ground materials in different situations. The similarity between these spectral signatures is very high, which leads to the conclusion that the solution of the sparse unmixed model is not unique. To overcome this disadvantage, we need to ensure that the linear correlation between the spectral characteristics in the spectral library is as small as possible. Therefore, we use a

strategy to precondition the spectral library. Given a spectral library $\mathbf{A}$, we calculate the spectral angle $\theta_{i,j}$ between any two different spectral characteristics $\mathbf{A}_i$ and $\mathbf{A}_j$. The $\theta_{i,j}$ is defined as follows:

$$\theta_{i,j} = \cos^{-1}\left(\frac{\mathbf{A}_i^T \mathbf{A}_j}{|\mathbf{A}_i||\mathbf{A}_j|}\right), \tag{12}$$

Any two spectral characteristics in spectral library $\mathbf{A}$ are regarded as linear correlation if the spectral angle $\theta_{i,j}$ of the two spectral characteristics is less than a given threshold. We reserve one of these linear correlation spectral characteristics and discard the remaining linear correlation spectral characteristics. With this operation, we obtain a pruned spectral library which is utilized for sparse unmixing.

### 3.2. Nonlocal Low-Rank Regularization

In the former section, we explained how the abundance matrix is converted to 3D form to find similar structural information. Let $\hat{\mathbf{X}} \in \mathbb{R}^{sr \times sc \times t}$ be the 3D form of the abundance data, where $sr$ and $sc$ represent the number of rows and columns, respectively, and $s = sr \times sc$ is the number of pixels in the abundance matrix and $t$ is the number of signatures in the spectral library. Assuming that the number of patches is $P$ in $\hat{\mathbf{X}}$ and $\hat{\mathbf{X}}_p \in \mathbb{R}^{sp \times sp \times tp}$ is the $p$-th patch, where $p = 1, 2, \ldots, P$. Taking the $p$-th patch as an example, we use a block matching algorithm [56] to find the $r$ nonlocal patches which are similar to the $p$-th patch. Then, with the $r+1$ patches together, we generate a patch group, which is denoted by $\hat{\mathbf{X}}_{r+1} \in \mathbb{R}^{sp \times sp \times u}$ and $u = tp \times (r+1)$. For each spectral dimension in the patch group, denoted by $\hat{\mathbf{X}}_{i,r+1} \in \mathbb{R}^{sp \times sp}$, where $i = 1, 2, \ldots, u$, we convert it to a vector, denoted by $\hat{\mathbf{x}}_{i,r+1} \in \mathbb{R}^w$, where $w = sp \times sp$. Figure 2 illustrates this process. Finally, we obtain the abundance matrix of the patch group by:

$$\mathbf{D}_{\hat{\mathbf{X}}_{r+1}} = (\hat{\mathbf{x}}_{1,r+1}, \hat{\mathbf{x}}_{2,r+1}, \ldots, \hat{\mathbf{x}}_{u,r+1}) \in \mathbb{R}^{u \times w} \tag{13}$$

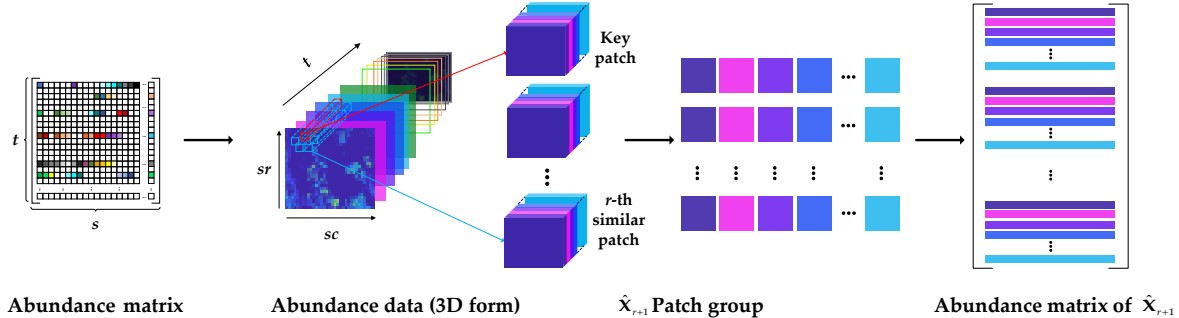

**Figure 2.** The process of obtaining the patch group and the nonlocal abundance matrix of the patch group. In the 3D form of abundance data, the small red box denotes a key patch, which slides across all dimensions in abundance data. The small blue boxes denote $r$ number of similar patches. Then we stack all $r+1$ patches to a patch group and convert this patch group into the vectors to obtain the abundance matrix.

The nuclear norm is then imposed to enforce the low-rank property of the abundance map $\mathbf{D}_{\hat{\mathbf{X}}_{r+1}}$ of the patch group $\hat{\mathbf{X}}_{r+1}$:

$$\|\mathbf{D}_{\hat{\mathbf{X}}_{r+1}}\|_* = \sum_{j=1}^{rank(\mathbf{D}_{\hat{\mathbf{X}}_{r+1}})} \sigma_j(\mathbf{D}_{\hat{\mathbf{X}}_{r+1}}), \tag{14}$$

where $\sigma_j$ represents the $j$-th singular value, where $j = 1, 2, \ldots, rank(\mathbf{D}_{\hat{\mathbf{X}}_{r+1}})$. Therefore, the estimated abundance matrix of this patch group can be enforced by the low-rank constraint. At last, we aggregate the estimated abundance matrix to the corresponding position of each patch in the patch group.

Similarly, we conduct this operation for all $P$ number of patches. For the whole abundance matrix $\mathbf{X}$, we use $\|\mathbf{X}\|_{NL*}$ to represent the nonlocal low-rank regularization term.

### 3.3. Proposed Model and Optimization

Based on the sparse unmixing model, the proposed NLLRSU algorithm has three regularizers, named collaborative sparsity, TV, and nonlocal abundance low-rankness. The model is formulated as follows:

$$\min_{\mathbf{X}} \frac{1}{2}\|\mathbf{AX} - \mathbf{Y}\|_F^2 + \lambda\|\mathbf{X}\|_{2,1} + \lambda_{TV}\|\mathbf{HX}\|_1 + \lambda_{NL}\|\mathbf{X}\|_{NL*} + l_{R+}(\mathbf{X}),  \tag{15}$$

where $\lambda, \lambda_{TV}, \lambda_{NL}$ are the parameters of the collaborative sparsity, TV, and nonlocal low-rank regularization terms, respectively. $l_{R+}(x)$ is an indicator function, when $x \geq 0$, $l_{R+}(x) = 0$; otherwise $l_{R+}(\mathbf{X}) = +\infty$, it is utilized to ensure ANC constraint.

It is difficult to optimize the model (15) directly, so we employ the ADMM [37] to solve it. The core idea of ADMM is that a difficult problem can be transformed into several simple subproblems by introducing some new variables cautiously. The subproblems are solved one by one and alternately updated. Here, we introduce the variables $\mathbf{V}_1, \mathbf{V}_2, \mathbf{V}_3, \mathbf{V}_4, \mathbf{V}_5, \mathbf{V}_6$. Then the Equation (15) can be rewritten as:

$$\min_{\mathbf{X},\mathbf{V}_1,\mathbf{V}_2,\mathbf{V}_3,\mathbf{V}_4,\mathbf{V}_5,\mathbf{V}_6} \frac{1}{2}\|\mathbf{V}_1 - \mathbf{Y}\|_F^2 + \lambda\|\mathbf{V}_2\|_{2,1} + \lambda_{TV}\|\mathbf{V}_4\|_1 + \lambda_{NL}\|\mathbf{V}_5\|_{NL*} + l_{R+}(\mathbf{V}_6)$$

$$\text{s.t.} \mathbf{V}_1 = \mathbf{AX}$$
$$\mathbf{V}_2 = \mathbf{X}$$
$$\mathbf{V}_3 = \mathbf{X}$$
$$\mathbf{V}_4 = \mathbf{HV}_3  \tag{16}$$
$$\mathbf{V}_5 = \mathbf{X}$$
$$\mathbf{V}_6 = \mathbf{X}$$

The Lagrangian function of the Equation (16) is:

$$\mathcal{L}(\mathbf{X}, \mathbf{V}_1, \mathbf{V}_2, \mathbf{V}_3, \mathbf{V}_4, \mathbf{V}_5, \mathbf{V}_6, \mathbf{D}_1, \mathbf{D}_2, \mathbf{D}_3, \mathbf{D}_4, \mathbf{D}_5, \mathbf{D}_6)$$
$$= \frac{1}{2}\|\mathbf{V}_1 - \mathbf{Y}\|_F^2 + \lambda\|\mathbf{V}_2\|_{2,1} + \lambda_{TV}\|\mathbf{V}_4\|_1 + \lambda_{NL}\|\mathbf{V}_5\|_{NL*}$$
$$+ l_{R+}(\mathbf{V}_6) + \frac{\mu}{2}\|\mathbf{V}_1 - \mathbf{AX} + \mathbf{D}_1\|_F^2 + \frac{\mu}{2}\|\mathbf{V}_2 - \mathbf{X} + \mathbf{D}_2\|_F^2  \tag{17}$$
$$+ \frac{\mu}{2}\|\mathbf{V}_3 - \mathbf{X} + \mathbf{D}_3\|_F^2 + \frac{\mu}{2}\|\mathbf{V}_4 - \mathbf{HV}_3 + \mathbf{D}_4\|_F^2$$
$$+ \frac{\mu}{2}\|\mathbf{V}_5 - \mathbf{X} + \mathbf{D}_5\|_F^2 + \frac{\mu}{2}\|\mathbf{V}_6 - \mathbf{X} + \mathbf{D}_6\|_F^2$$

where $\mathbf{D}_1, \mathbf{D}_2, \mathbf{D}_3, \mathbf{D}_4, \mathbf{D}_5, \mathbf{D}_6$ are Lagrangian multipliers and $\mu$ is the Lagrangian parameter.

Algorithm 1 gives the pseudocode for the NLLRSU solution process. In each iteration, with the ADMM, we optimize $\mathbf{X}, \mathbf{V}_1, \mathbf{V}_2, \mathbf{V}_3, \mathbf{V}_4, \mathbf{V}_5, \mathbf{V}_6$ in sequence, then we update the Lagrangian multipliers $\mathbf{D}_1, \mathbf{D}_2, \mathbf{D}_3, \mathbf{D}_4, \mathbf{D}_5, \mathbf{D}_6$.

Next, we discuss the details of step 3 in Algorithm 1, which is utilized to compute the value of variable $\mathbf{X}$ at each iteration. In this step, we fix other variables and update $\mathbf{X}$ only. The optimization problem for step 3 can be written as:

$$\mathbf{X}^{(k+1)} \leftarrow \arg\min_{\mathbf{X}} \frac{\mu}{2}\|\mathbf{V}_1^{(k)} - \mathbf{AX} + \mathbf{D}_1^{(k)}\|_F^2 + \frac{\mu}{2}\|\mathbf{V}_2^{(k)} - \mathbf{X} + \mathbf{D}_2^{(k)}\|_F^2$$
$$+ \frac{\mu}{2}\|\mathbf{V}_3^{(k)} - \mathbf{X} + \mathbf{D}_3^{(k)}\|_F^2 + \frac{\mu}{2}\|\mathbf{V}_5^{(k)} - \mathbf{X} + \mathbf{D}_5^{(k)}\|_F^2 + \frac{\mu}{2}\|\mathbf{V}_6^{(k)} - \mathbf{X} + \mathbf{D}_6^{(k)}\|_F^2  \tag{18}$$

The solution of Equation (18) is:

$$\mathbf{X}^{(k+1)} \leftarrow (\mathbf{A}^T\mathbf{A} + 4\mathbf{I})^{-1}(\mathbf{A}^T\xi_1 + \xi_2 + \xi_3 + \xi_5 + \xi_6)  \tag{19}$$

where $\mathbf{A}^T$ is the transpose of $\mathbf{A}$, $\mathbf{I}$ is the identity matrix, and $\xi_1 = \mathbf{V}_1^{(k)} + \mathbf{D}_1^{(k)}, \xi_2 = \mathbf{V}_2^{(k)} + \mathbf{D}_2^{(k)}$, $\xi_3 = \mathbf{V}_3^{(k)} + \mathbf{D}_3^{(k)}$, $\xi_5 = \mathbf{V}_5^{(k)} + \mathbf{D}_5^{(k)}$, $\xi_6 = \mathbf{V}_6^{(k)} + \mathbf{D}_6^{(k)}$.

---

**Algorithm 1:** Pseudocode of the NLLRSU algorithm.

---

1. Initialization: set $k = 0$, choose $\mu, \lambda, \lambda_{TV}, \lambda_{NL}, \mathbf{X}^{(0)}, \mathbf{V}_1^{(0)}, \ldots, \mathbf{V}_6^{(0)}, \mathbf{D}_1^{(0)}, \ldots, \mathbf{D}_6^{(0)}$
2. while some stopping criterion is not satisfied do
3. $\mathbf{X}^{(k+1)} \leftarrow \underset{\mathbf{X}}{\mathrm{argmin}} \mathcal{L}(\mathbf{X}, \mathbf{V}_1^{(k)}, \ldots, \mathbf{V}_6^{(k)}, \mathbf{D}_1^{(k)}, \ldots, \mathbf{D}_6^{(k)})$
4. for $i = 1, \ldots, 6$ do
5. $\mathbf{V}_i^{(k+1)} \leftarrow \underset{\mathbf{V}_i}{\mathrm{argmin}} \mathcal{L}(\mathbf{X}^{(k)}, \mathbf{V}_1^{(k)}, \ldots, \mathbf{V}_i, \ldots, \mathbf{V}_6^{(k)})$
6. end for
7. Update Lagrange multipliers
8. $\mathbf{D}_1^{(k+1)} \leftarrow \mathbf{D}_1^{(k)} - \mathbf{A}\mathbf{X}^{(k+1)} + \mathbf{V}_1^{(k+1)}$
9. $\mathbf{D}_4^{(k+1)} \leftarrow \mathbf{D}_4^{(k)} - \mathbf{H}\mathbf{V}_3^{(k+1)} + \mathbf{V}_4^{(k+1)}$
10. $\mathbf{D}_i^{(k+1)} \leftarrow \mathbf{D}_i^{(k)} - \mathbf{X}^{(k+1)} + \mathbf{V}_i^{(k+1)}, i = 2, 3, 5, 6$
11. Update iteration $k = k + 1$
12. end while

---

Now, we introduce solutions of the variables $\mathbf{V}_1, \mathbf{V}_2, \mathbf{V}_3, \mathbf{V}_4, \mathbf{V}_5, \mathbf{V}_6$ in step 5 of Algorithm 1. To solve $\mathbf{V}_1$, the optimization problem for $\mathbf{V}_1$ can be described as:

$$\mathbf{V}_1^{(k+1)} \leftarrow \underset{\mathbf{V}_1}{\mathrm{argmin}} \frac{1}{2}\|\mathbf{V}_1 - \mathbf{Y}\|_F^2 + \frac{\mu}{2}\|\mathbf{V}_1 - \mathbf{A}\mathbf{X}^{(k)} + \mathbf{D}_1^{(k)}\|_F^2 \tag{20}$$

The solution of $\mathbf{V}_1$ is:

$$\mathbf{V}_1^{(k+1)} \leftarrow \frac{1}{1 + \mu}[\mathbf{Y} + \mu(\mathbf{A}\mathbf{X}^{(k)} - \mathbf{D}_1^{(k)})] \tag{21}$$

The optimization problem of $\mathbf{V}_2$ is:

$$\mathbf{V}_2^{(k+1)} \leftarrow \underset{\mathbf{V}_2}{\mathrm{argmin}} \lambda\|\mathbf{V}_2\|_{2,1} + \frac{\mu}{2}\|\mathbf{V}_2 - \mathbf{X}^{(k)} + \mathbf{D}_2^{(k)}\|_F^2 \tag{22}$$

The solution of $\mathbf{V}_2$ is given by using the famous vect-soft threshold [62]:

$$\mathbf{V}_2^{(k+1)} \leftarrow \text{vect-soft}(\mathbf{X}^{(k)} - \mathbf{D}_2^{(k)}, \frac{\lambda}{\mu}) \tag{23}$$

where vect-soft$(\cdot, \psi)$ means that the vect-soft threshold function is applied row by row, and the vect-soft threshold function is $x \mapsto x(\max\{\|x\|_2 - \psi, 0\} / \max\{\|x\|_2 - \psi, 0\} + \psi)$, $\psi$ is a threshold.

To compute $\mathbf{V}_3$, we need to solve the following problem:

$$\mathbf{V}_3^{(k+1)} \leftarrow \underset{\mathbf{V}_3}{\mathrm{argmin}} \frac{\mu}{2}\|\mathbf{V}_3 - \mathbf{X}^{(k)} + \mathbf{D}_3^{(k)}\|_F^2 + \frac{\mu}{2}\|\mathbf{V}_4^{(k)} - \mathbf{H}\mathbf{V}_3 + \mathbf{D}_4^{(k)}\|_F^2 \tag{24}$$

The solution of $\mathbf{V}_3$ is:

$$\mathbf{V}_3^{(k+1)} \leftarrow (\mathbf{H}^T\mathbf{H} + \mathbf{I})^{-1}(\mathbf{X}^{(k)} - \mathbf{D}_3^{(k)} + \mathbf{H}^T(\mathbf{V}_4^{(k)} + \mathbf{D}_4^{(k)})) \tag{25}$$

where **H** represents a convolution. **H** can be applied independently in a band-by-band manner and calculated by discrete Fourier transform diagonalization [63].

The optimization problem of $\mathbf{V}_4$ is:

$$\mathbf{V}_4^{(k+1)} \leftarrow \underset{\mathbf{V}_4}{\operatorname{argmin}} \lambda_{TV} \|\mathbf{V}_4\|_1 + \frac{\mu}{2} \|\mathbf{V}_4 - \mathbf{H}\mathbf{V}_3^{(k)} + \mathbf{D}_4^{(k)}\|_F^2 \tag{26}$$

The solution of $\mathbf{V}_4$ can be obtained by soft-threshold [64]:

$$\mathbf{V}_4^{(k+1)} \leftarrow \operatorname{soft}(\mathbf{D}_4^{(k)} - \mathbf{H}\mathbf{V}_3^{(k)}, \frac{\lambda_{TV}}{\mu}) \tag{27}$$

where $\operatorname{soft}(\cdot, \psi)$ represents the soft-threshold function $x \mapsto \operatorname{sign}(x)\max\{|x| - \psi, 0\}$.

To obtain the solution of $\mathbf{V}_5$, the optimization problem of (28) should be solved.

$$\mathbf{V}_5^{(k+1)} \leftarrow \underset{\mathbf{V}_5}{\operatorname{argmin}} \lambda_{NL} \|\mathbf{V}_5\|_{NL*} + \frac{\mu}{2} \|\mathbf{V}_5 - \mathbf{X}^{(k)} + \mathbf{D}_5^{(k)}\|_F^2 \tag{28}$$

For the abundance matrix $\mathbf{D}_{\hat{\mathbf{X}}_{r+1}}$ of each patch group, we apply singular value shrinkage to obtain the reconstructed abundance matrix. The solution of $\mathbf{V}_5$ can be expressed as:

$$\mathbf{V}_5^{(k+1)} \leftarrow \operatorname{shrinkage}\left(\mathbf{X}^{(k)} - \mathbf{D}_5^{(k)}, \frac{\lambda_{NL}}{\mu}\right) \tag{29}$$

where $\operatorname{shrinkage}(\cdot, \psi)$ is the singular value shrinkage $x \mapsto \operatorname{diag}(\max\{SVD(x) - \psi, 0\})$ of abundance matrix $\mathbf{D}_{\hat{\mathbf{X}}_{r+1}}$.

Finally, the optimization problem of $\mathbf{V}_6$ is as follows:

$$\mathbf{V}_6^{(k+1)} \leftarrow \underset{\mathbf{V}_6}{\operatorname{argmin}} l_{R+}(\mathbf{V}_6) + \frac{\mu}{2} \|\mathbf{V}_6 - \mathbf{X}^{(k)} + \mathbf{D}_6^{(k)}\|_F^2 \tag{30}$$

The solution of $\mathbf{V}_6$ is:

$$\mathbf{V}_6^{(k+1)} \leftarrow \max\left(\mathbf{X}^{(k)} - \mathbf{D}_6^{(k)}, 0\right) \tag{31}$$

### 3.4. Computational Complexity

In this algorithm, the most time-consuming procedures are calculating $\mathbf{X}$, $\mathbf{V}_3$, and $\mathbf{V}_5$, and the corresponding complexities are $O(l^2 s)$, $O(ls\log s)$, and $O(wu^2 P)$, respectively, where $s$ is the total number of pixels in a hyperspectral image, $l$ is the number of spectral characteristics, $w$ and $u$ are the number of pixels and bands in one patch group, respectively, and $P$ is the number of patches. In each iteration, $\mathbf{V}_5$, the singular value decomposition step, costs the most time. Thus, the overall complexity of this algorithm is $O(wu^2 P)$.

## 4. Experiment and Analysis

In order to test the unmixing performance of the proposed algorithm, we used two simulated data sets and two real hyperspectral image data sets. For each simulated data set, our experiments were performed in three different signal-to-noise ratio (SNR) levels, 10dB, 15dB, and 20dB. We also compared the proposed algorithm with SUnSAL [31], CLSUnSAL [38], SUnSAL-TV [42], J-LASU [55], and $L_1$-$L_2$ SUnSAL-TV [43].

### 4.1. Experiments with Simulated Data

In this paper, we employed splib06 [65], a USGS spectral library released in September 2007, to our experiments. We selected 240 endmember signatures randomly from the splib06 as the spectral

library, denoted by $\mathbf{A} \in \mathbb{R}^{224 \times 240}$, and the spectral library $\mathbf{A}$ contains 224 bands of spectra with spectral range from 0.4 to 2.5 μm. Because the linear correlation between the spectral characteristics is very high, we set the spectral angle to be larger than 4.4 to avoid this problem.

The simulated data set 1 (DS1) was generated by randomly selecting five spectral characteristics from the spectral library $\mathbf{A}$ and following the linear mixed model. DS1 has $75 \times 75$ pixels and each pixel has 224 bands and the corresponding abundance is constrained by ASC. Figure 3a illustrated the composition of the DS1. In DS1, some square regions are pure and some regions are mixed by two to five endmembers. The background pixels of this data set are composed of the same five endmembers with randomly fixed abundance values 0.1149, 0.0741, 0.2003, 0.2055, and 0.4051. Figure 3b–f shows the true abundances of five endmembers. After generating DS1, we applied three different SNR levels Gaussian noise, 10dB, 15dB, and 20dB, to this data set.

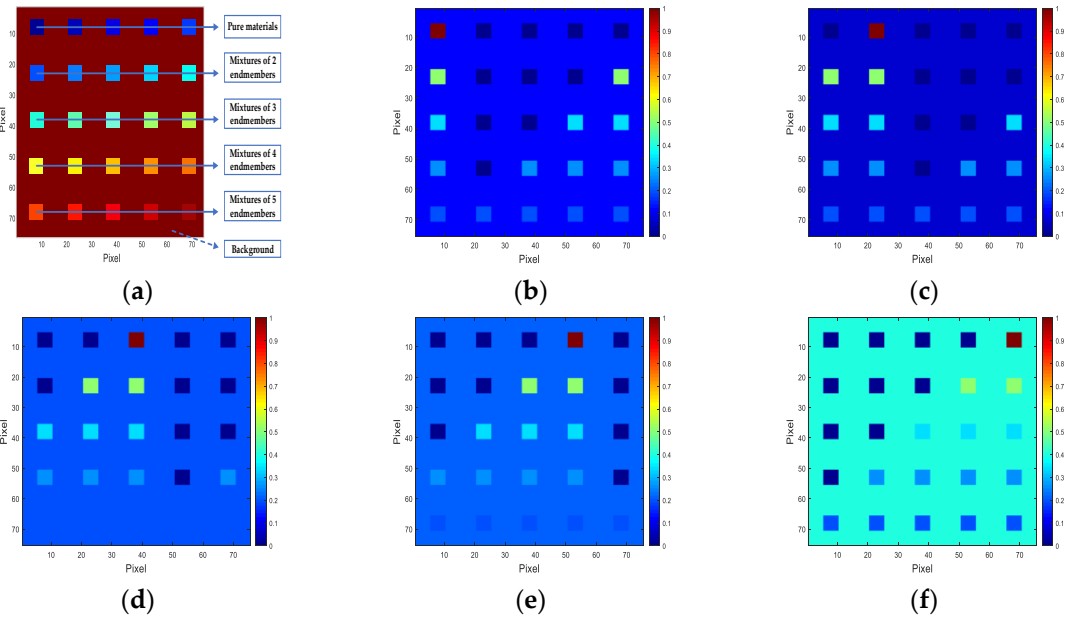

**Figure 3.** True fractional abundances of endmembers in the simulated data set 1 (DS1). (**a**) Simulated image; (**b**) endmember 1; (**c**) endmember 2; (**d**) endmember 3; (**e**) endmember 4; (**f**) endmember 5.

The simulated data set 2 (DS2) contains nine spectral characteristics which are randomly selected from spectral library $\mathbf{A}$ and has $100 \times 100$ pixels. The abundances of nine spectral signatures obey a Dirichlet distribution uniformly over the probability simplex and are constrained by ANC and ASC. As shown in Figure 4, the abundances of DS2 shows the piecewise smoothness in the spatial domain and their distributions are closer to those of the real data sets. After generating DS2, this data set is distorted by Gaussian noise with the same SNR levels as DS1.

In the simulated data experiment, the parameters $\lambda$ represent sparsity term for SUnSAL, CLSUnSAL, SUnSAL-TV, J-LASU, $L_1$-$L_2$ SUnSAL-TV, and NLLRSU. The TV term for SUnSAL-TV, J-LASU, $L_1$-$L_2$ SUnSAL-TV, and NLLRSU is represented by $\lambda_{TV}$. For J-LASU, the local low-rank term is controlled by $\lambda_{LR}$ and the local block size is set to $5 \times 5 \times 5$ with no overlap according to J-LASU. The $\lambda_{NL}$ means nonlocal abundance low-rank term for NLLRSU and each local patch size is set to $5 \times 5 \times 5$ and we find the four patches which are similar to the local patch after several experiments. The parameters $\lambda$, $\lambda_{TV}$, $\lambda_{LR}$, $\lambda_{NL}$ are selected between $10^{-4}$ to 10. For each algorithm, we adjust each parameter to the optimal value. Table 1 shows these parameter settings.

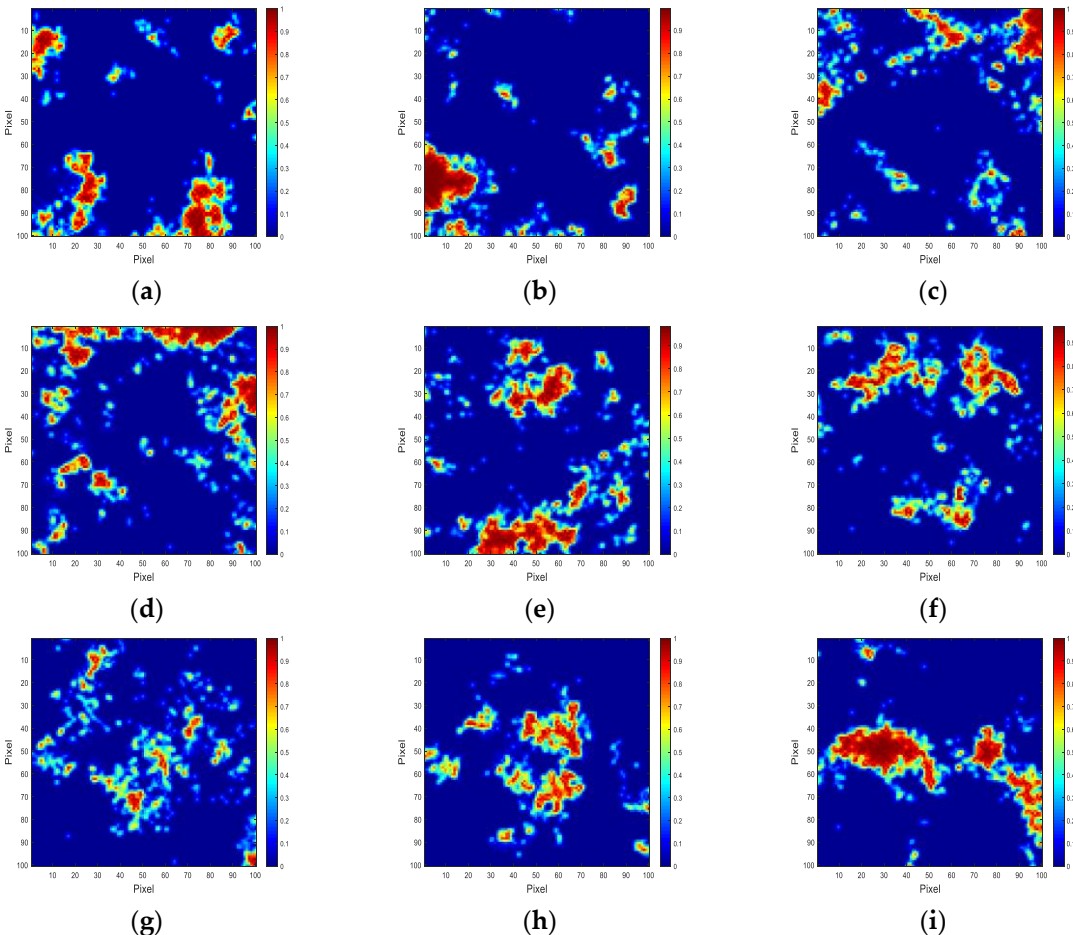

**Figure 4.** True fractional abundances of endmembers in the simulated data set 2 (DS2). (**a**) Endmember 1; (**b**) endmember 2; (**c**) endmember 3; (**d**) endmember 4; (**e**) endmember 5; (**f**) endmember 6; (**g**) endmember 7; (**h**) endmember 8; (**i**) endmember 9.

**Table 1.** Parameter settings.

| Data | | DS1 | | | DS2 | | |
|---|---|---|---|---|---|---|---|
| **SNR** | | **10dB** | **15dB** | **20dB** | **10dB** | **15dB** | **20dB** |
| **SUnSAL** [31] | $\lambda$ | $1 \times 10^0$ | $5 \times 10^{-1}$ | $1 \times 10^{-1}$ | $5 \times 10^{-1}$ | $1 \times 10^{-1}$ | $1 \times 10^{-1}$ |
| **CLSUnSAL** [38] | $\lambda$ | $1 \times 10^1$ | $1 \times 10^1$ | $8 \times 10^0$ | $1 \times 10^1$ | $1 \times 10^1$ | $3 \times 10^0$ |
| **SUnSAL-TV** [42] | $\lambda$ | $1 \times 10^{-1}$ | $1 \times 10^{-1}$ | $5 \times 10^{-2}$ | $1 \times 10^{-1}$ | $1 \times 10^{-1}$ | $5 \times 10^{-2}$ |
| | $\lambda_{TV}$ | $5 \times 10^{-1}$ | $1 \times 10^{-1}$ | $5 \times 10^{-2}$ | $1 \times 10^{-1}$ | $5 \times 10^{-2}$ | $1 \times 10^{-2}$ |
| **J-LASU** [55] | $\lambda$ | $2 \times 10^0$ | $1 \times 10^0$ | $2.5 \times 10^{-1}$ | $1 \times 10^{-4}$ | $1 \times 10^{-4}$ | $1 \times 10^{-4}$ |
| | $\lambda_{TV}$ | $1 \times 10^{-1}$ | $1 \times 10^{-1}$ | $5 \times 10^{-2}$ | $5 \times 10^{-2}$ | $5 \times 10^{-2}$ | $1 \times 10^{-2}$ |
| | $\lambda_{LR}$ | $1 \times 10^0$ | $5 \times 10^{-1}$ | $3 \times 10^{-1}$ | $5 \times 10^{-1}$ | $1 \times 10^{-1}$ | $1 \times 10^{-1}$ |
| $L_1$-$L_2$ **SUnSAL-TV** [43] | $\lambda$ | $1 \times 10^{-2}$ | $1 \times 10^{-1}$ | $1 \times 10^{-2}$ | $1 \times 10^{-1}$ | $1 \times 10^{-2}$ | $1 \times 10^{-2}$ |
| | $\lambda_{TV}$ | $5 \times 10^{-1}$ | $5 \times 10^{-2}$ | $5 \times 10^{-2}$ | $1 \times 10^{-1}$ | $5 \times 10^{-2}$ | $5 \times 10^{-2}$ |
| **NLLRSU** | $\lambda$ | $5 \times 10^{-1}$ | $1 \times 10^{-1}$ | $1 \times 10^{-1}$ | $1 \times 10^{-4}$ | $1 \times 10^{-4}$ | $1 \times 10^{-4}$ |
| | $\lambda_{TV}$ | $1 \times 10^{-1}$ | $1 \times 10^{-1}$ | $5 \times 10^{-2}$ | $1 \times 10^{-1}$ | $5 \times 10^{-2}$ | $1 \times 10^{-2}$ |
| | $\lambda_{NL}$ | $2 \times 10^0$ | $5 \times 10^{-1}$ | $5 \times 10^{-1}$ | $1 \times 10^0$ | $5 \times 10^{-1}$ | $1 \times 10^{-1}$ |

We employ two evaluation indicators to assess the quality of the six algorithms: signal reconstruction error (SRE) [66] and root mean square error (RMSE) [67]. SRE is the ratio between the reconstructed abundance matrix and error, and higher SRE value indicates a better result. SRE can be defined as follows:

$$\mathrm{SRE(dB)} = 10\log_{10}\frac{\mathrm{E}[\|\mathbf{X}\|_2^2]}{\mathrm{E}[\|\mathbf{X}-\hat{\mathbf{X}}\|_2^2]}, \tag{32}$$

where $\mathbf{X}$ is the true abundance matrix and $\hat{\mathbf{X}}$ is the estimated abundance matrix.

RMSE represents the error between the true abundance matrix and the estimated abundance matrix, and lower RMSE value indicates more accurate estimation of abundance matrix. RMSE can be defined as follows:

$$\mathrm{RMSE} = \sqrt{\frac{1}{t\times s}\sum_{i=1}^{t}\sum_{j=1}^{s}(x_{ij}-\hat{x}_{ij})^2}, \tag{33}$$

where $x_{ij}$ and $\hat{x}_{ij}$ represent each element in the true abundance matrix and the estimated abundance matrix, respectively.

Figure 5 shows SRE(dB) values of NLLRSU as a function of parameters $\lambda$, $\lambda_{TV}$, and $\lambda_{NL}$ in 20dB SNR level. Since there are three parameters, we considering the effect of ($\lambda$ and $\lambda_{TV}$), ($\lambda$ and $\lambda_{NL}$), ($\lambda_{TV}$ and $\lambda_{NL}$) on SRE value, respectively, while the other parameters are set to the optimal values. From Figure 5, we can observe that relatively smaller parameter values result in better SRE values.

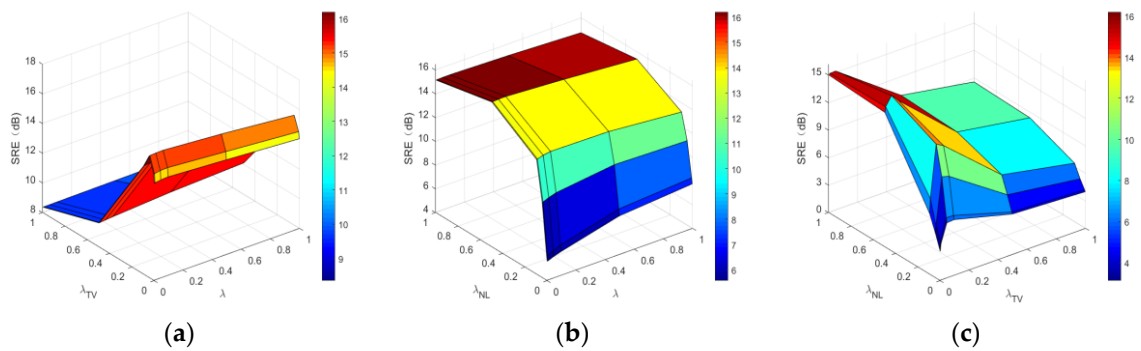

(a)             (b)             (c)

**Figure 5.** Signal reconstruction error (SRE)(dB) as a function of parameters $\lambda$, $\lambda_{TV}$, and $\lambda_{NL}$ for DS1 with 20dB SNR level. (**a**) $\lambda$ and $\lambda_{TV}$; (**b**) $\lambda$ and $\lambda_{NL}$; (**c**) $\lambda_{TV}$ and $\lambda_{NL}$.

Figures 6–9 show the abundance image reconstructed by the six algorithms for one randomly selected endmember in DS1 and DS2 with 20db SNR level, respectively. From 8 and 9, we can clearly see that SUnSAL-TV algorithm shows the smoothest results in the edge regions. Nevertheless, the edge transition regions reconstructed by our algorithm is closer to the true abundance distribution. The result shows that nonlocal low-rank regularization terms keep the structure information of the image better than other algorithms. Tables 2 and 3 show the values of SRE (dB) and RMSE of the six algorithms with two simulated data sets, respectively. From these tables, we can observe that the performance of the proposed NLLRSU algorithm is better than other state-of-the-art algorithms in the same SNR level.

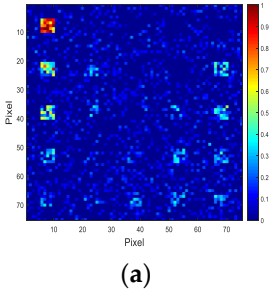     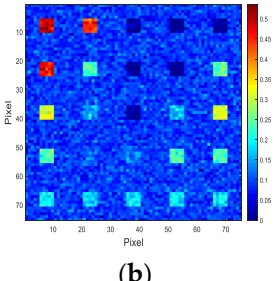     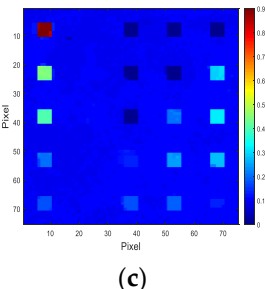

(a)             (b)             (c)

**Figure 6.** *Cont.*

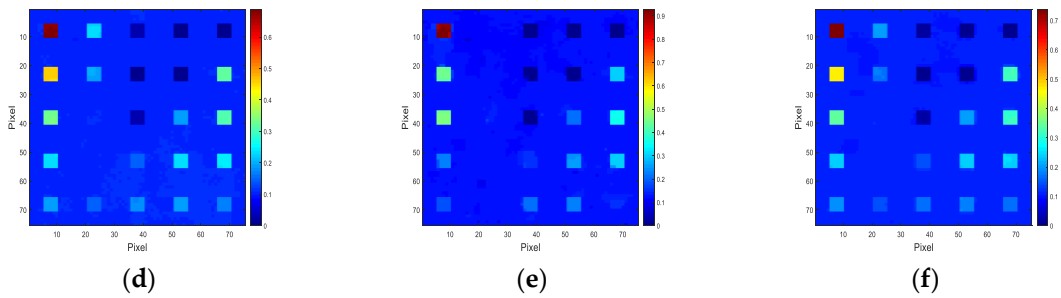

**Figure 6.** Reconstructed fractional abundances of endmember 1 in DS1 with 20 dB SNR level. (**a**) SUnSAL; (**b**) CLSUnSAL; (**c**) SUnSAL-TV; (**d**) J-LASU; (**e**) $L_1$-$L_2$ SUnSAL-TV; (**f**) NLLRSU.

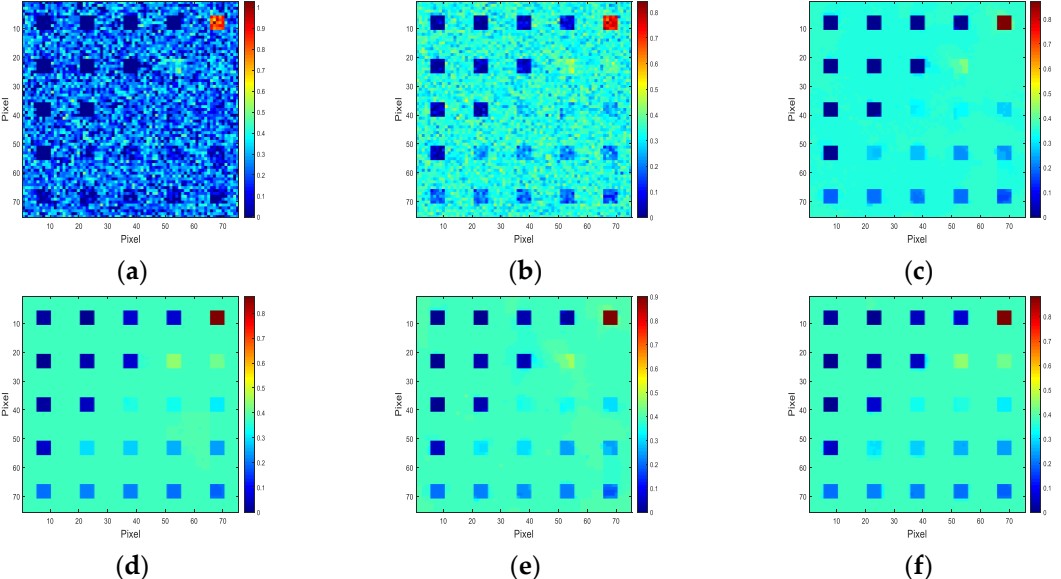

**Figure 7.** Reconstructed fractional abundances of endmember 5 in DS1 with 20 dB SNR level. (**a**) SUnSAL; (**b**) CLSUnSAL; (**c**) SUnSAL-TV; (**d**) J-LASU; (**e**) $L_1$-$L_2$ SUnSAL-TV; (**f**) NLLRSU.

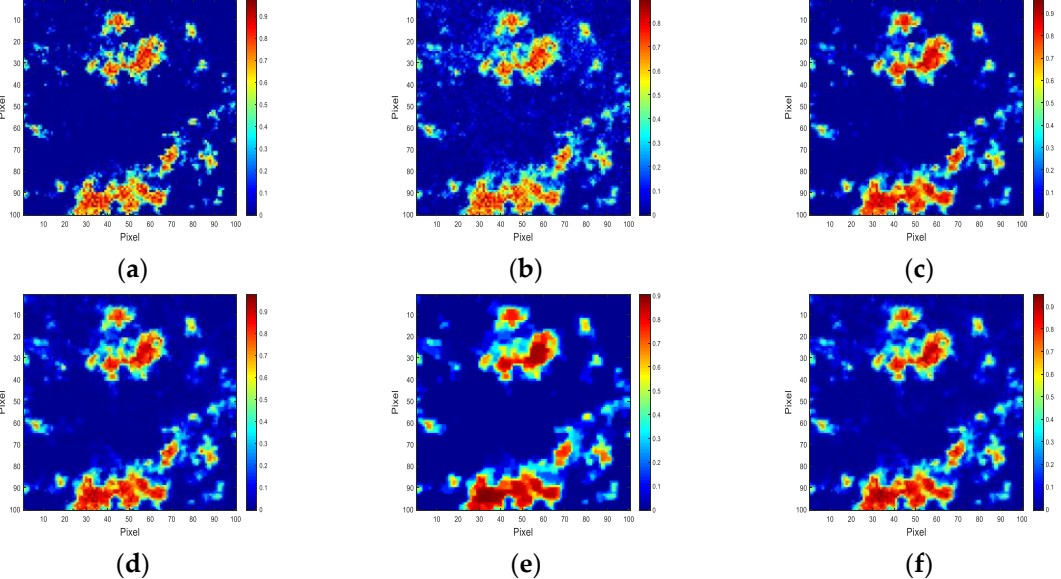

**Figure 8.** Reconstructed fractional abundances of endmember 5 in DS2 with 20 dB SNR level. (**a**) SUnSAL; (**b**) CLSUnSAL; (**c**) SUnSAL-TV; (**d**) J-LASU; (**e**) $L_1$-$L_2$ SUnSAL-TV; (**f**) NLLRSU.

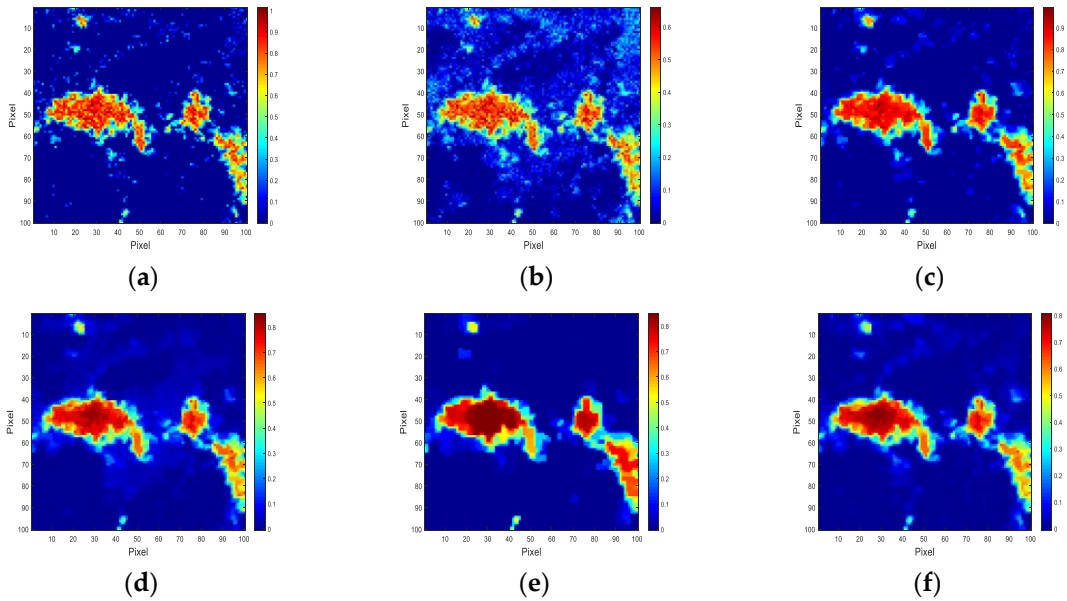

**Figure 9.** Reconstructed fractional abundances of endmember 9 in DS2 with 20 dB SNR level. (**a**) SUnSAL; (**b**) CLSUnSAL; (**c**) SUnSAL-TV; (**d**) J-LASU; (**e**) $L_1$-$L_2$ SUnSAL-TV; (**f**) NLLRSU.

**Table 2.** SRE (dB) result (The optimal results are shown in bold type).

| Data | SNR | SUnSAL [31] | CLSUnSAL [38] | SUnSAL-TV [42] | J-LASU [55] | $L_1$-$L_2$ SUnSAL-TV [43] | NLLRSU |
|------|-----|-------------|---------------|----------------|-------------|---------------------------|--------|
| DS1 | 10dB | 0.2017 | 1.6544 | 3.9803 | 10.6039 | 4.1424 | **11.5466** |
|     | 15dB | 0.9448 | 4.4945 | 6.4204 | 13.4001 | 6.0267 | **13.9307** |
|     | 20dB | 2.4218 | 6.0518 | 7.1069 | 15.3860 | 7.0245 | **16.2008** |
| DS2 | 10dB | 1.2568 | 1.0144 | 3.6093 | 3.5925 | 3.6494 | **3.9300** |
|     | 15dB | 1.9727 | 2.3747 | 4.5860 | 4.8039 | 4.8445 | **5.1034** |
|     | 20dB | 4.1627 | 3.3961 | 5.5352 | 6.3062 | 6.2078 | **6.7188** |

**Table 3.** Root mean square error (RMSE) result (The optimal results are shown in bold type).

| Data | SNR | SUnSAL [31] | CLSUnSAL [38] | SUnSAL-TV [42] | J-LASU [55] | $L_1$-$L_2$ SUnSAL-TV [43] | NLLRSU |
|------|-----|-------------|---------------|----------------|-------------|---------------------------|--------|
| DS1 | 10dB | 0.0338 | 0.0286 | 0.0218 | 0.0102 | 0.0214 | **0.0091** |
|     | 15dB | 0.0310 | 0.0206 | 0.0165 | 0.0074 | 0.0173 | **0.0069** |
|     | 20dB | 0.0261 | 0.0172 | 0.0152 | 0.0059 | 0.0154 | **0.0054** |
| DS2 | 10dB | 0.0426 | 0.0438 | 0.0325 | 0.0325 | 0.0323 | **0.0313** |
|     | 15dB | 0.0392 | 0.0374 | 0.0290 | 0.0283 | 0.0282 | **0.0273** |
|     | 20dB | 0.0305 | 0.0333 | 0.0260 | 0.0238 | 0.0241 | **0.0227** |

### 4.2. Experiments with Real Data

For the real hyperspectral data set experiment, we utilized the famous AVIRIS Cuprite mine map and Urban data to test the performance of the six algorithms. For the Cuprite mine map, we utilized a subset of this data set, with 250 × 191 pixels. This subset contains 224 spectral bands between 0.4–2.5 μm. Because of the low SNR and water absorption, we removed the bands of 1-2, 105-115, 150-170, 223-224, and left 188 bands. The sub-library of USGS with 50 signatures was used to unmix this subset. Figure 10 shows the USGS mineral distribution of Cuprite in Nevada. This mineral map was drawn by Tricorder 3.3 software in 1995. However, the data for the currently published AVIRIS Cuprite mine was collected in 1997. Therefore, this mineral map was used for qualitative assessment of the six algorithms, because the mineral map product in 1995 and the AVIRIS data collected in 1997 cannot be directly compared. The Urban data has 307 × 307 pixels and 210 spectral bands with a range of wavelengths between 0.4 and 2.5 μm. We utilized a subset with 100 × 100 pixels for the experiment.

The bands 1–4, 76, 87, 101–111, 136–153, and 198-210 have been removed because of atmospheric influence and water absorption, leaving 162 spectral bands. Since the ground truth of the Urban data set is not available, we use the abundance map obtained from [68] as the ground truth, which is obtained by the method of [69,70], including four endmembers (asphalt road, grass, tree, and roof).

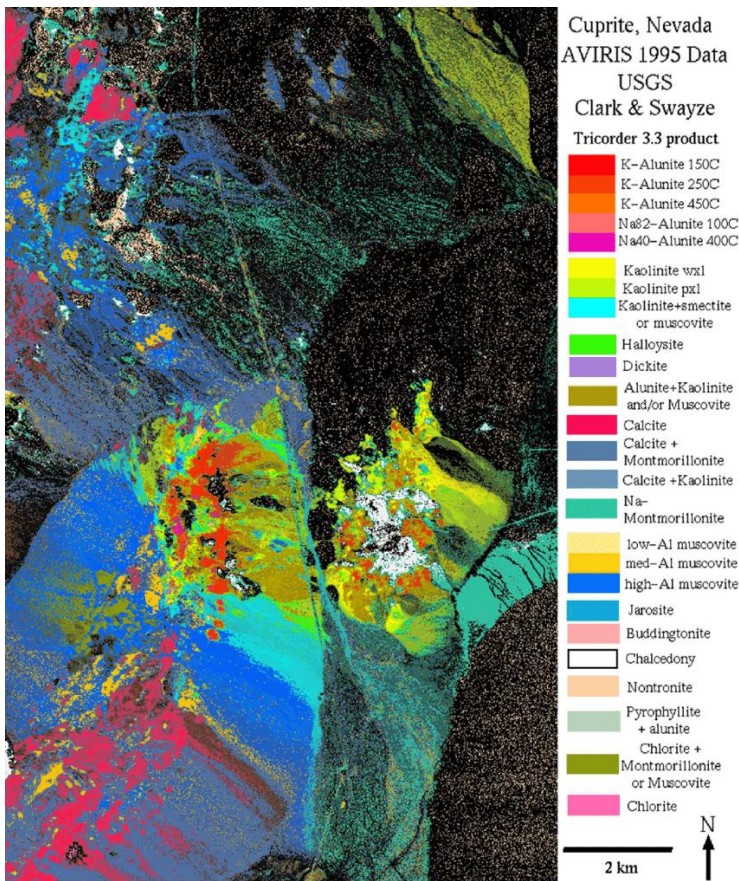

**Figure 10.** United States Geological Survey (USGS) mineral map of Cuprite in Nevada.

Figure 11 shows the three minerals' (Alunite, Buddingtonite and Chalcedony) abundance reconstructed by SUnSAL, CLSUnSAL, SUnSAL-TV, J-LASU, $L_1$-$L_2$ SUnSAL-TV, and NLLRSU algorithms. In Figure 11, the first row shows the mineral distribution of Alunite, Buddingtonite, and Chalcedony, which was generated by Tricorder 3.3 software in 1995 and regarded as ground truth. Although it is difficult to perform the qualitative assessment of the results from the real data produced by these algorithms, we can directly observe the results of the unmixing from the reconstructed abundance map. From Figure 11, it is clear to see that SUnSAL and CLSUnSAL perform poorly in the aspect of noise suppression while SUnSAL-TV, J-LASU, $L_1$-$L_2$ SUnSAL-TV, and NLLRSU produce smooth transition results in the edge regions because the methods with TV regularization and NLLRSU term keep structure information better than other algorithms. Table 4 shows the SRE and RMSE results of the six algorithms on Urban data. Since the ground truth of Urban data cannot be obtained directly from ground measurement but obtained by an algorithm, the RMSE values of the six algorithms are much higher than those from the simulated data sets. From Table 4, it is obvious that the unmixing results obtained by the proposed algorithm are better than other state-of-the-art algorithms. The experiments on real datasets show that the nonlocal low-rank term can improve the unmixing results effectively.

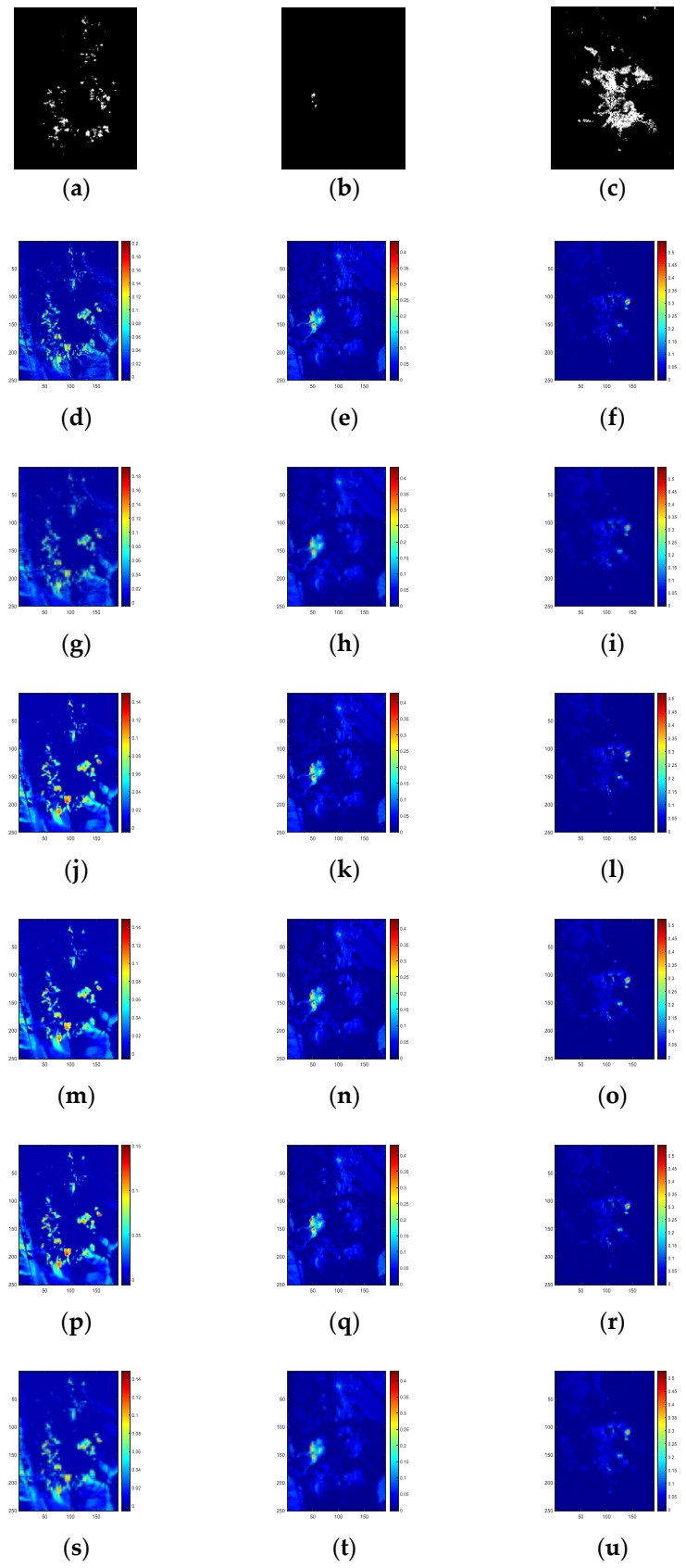

**Figure 11.** The first row (**a–c**) shows the distribution maps of Alunite, Buddingtonite, and Chalcedony (column 1-3) by Tricorder software. The second row (**d–u**) to the seventh row shows the reconstructed abundances maps of three minerals by SUnSAL, CLSUnSAL, SUnSAL-TV, J-LASU, $L_1$-$L_2$ SUnSAL-TV, NLLRSU.

**Table 4.** SRE and RMSE results of Urban data (The optimal results are shown in bold type).

| Algorithms | SUnSAL [31] | CLSUnSAL [38] | SUnSAL-TV [42] | J-LASU [55] | $L_1$-$L_2$ SUnSAL-TV [43] | NLLRSU |
|---|---|---|---|---|---|---|
| SRE | 3.9422 | 4.1685 | 4.2535 | 4.6928 | 4.5506 | **4.9362** |
| RMSE | 0.5534 | 0.5392 | 0.5339 | 0.5076 | 0.5160 | **0.4935** |

*4.3. Discussion*

Except for the parameters, the patch size and patch number also affect the optimal results. In each patch group consisting of nonlocal patches, we utilized the nuclear norm to exploit its low-rank property. We performed several experiments to find the optimal patch size and patch number in DS1 and DS2. Figure 12a shows the effect of patch size and patch number on SRE in DS1. Figure 12b shows SRE and RMSE results as a function of patch size and patch number in DS1. After several experiments, we found the optimal patch size is $5 \times 5$ pixels with 5 spectral bands and the patch number is 5. Therefore, we utilized $5 \times 5 \times 5$ patch size and 5 patch number for all data sets.

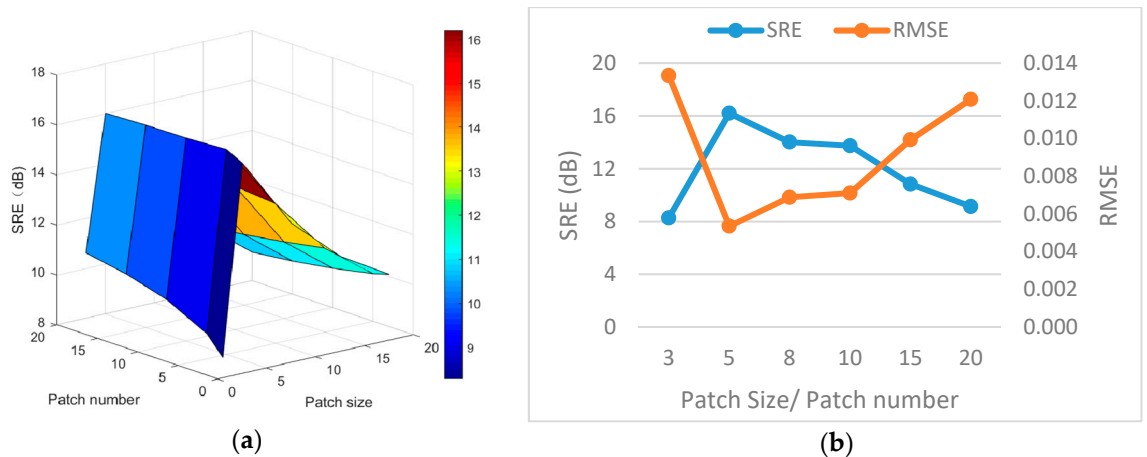

**Figure 12.** (**a**) Effect of patch size and number on SRE; (**b**) SRE and RMSE result with patch size and patch number change.

We conducted the experiments on the DS1 and the Cuprite mine map data set to compare the running times of the six algorithms. DS1 has $75 \times 75$ pixels with 224 spectral bands and Cuprite mine map data set has $250 \times 191$ pixels with 188 spectral bands. The six algorithms were performed on Intel Core i7 with 3.6 GHz and 8 GB RAM desktop computer. Table 5 shows the execution time of the six algorithms. It is clear to see that the proposed algorithm (NLLRSU) takes the longest time because our method is the most complex one among the six algorithms. However, the proposed algorithm preserves the details of the abundance map best, especially when the noise level is high.

**Table 5.** Execution time comparison (seconds/iteration).

| Data | SUnSAL [31] | CLSUnSAL [38] | SUnSAL-TV [42] | J-LASU [55] | $L_1$-$L_2$ SUnSAL-TV [43] | NLLRSU |
|---|---|---|---|---|---|---|
| DS1 | 0.08 | 0.09 | 0.33 | 1.51 | 0.34 | 2.89 |
| AVIRIS data | 0.21 | 0.22 | 0.62 | 2.87 | 0.66 | 5.76 |

## 5. Conclusions

In this paper, a nonlocal low-rank sparse unmixing algorithm which explores both spectral and spatial information is proposed to improve the unmixing performance. In this method, a novel nonlocal low-rank regularization, a collaborative term and a TV term are integrated into a unified framework.

The proposed model is finally solved by the ADMM algorithm effectively. Two simulated data sets and two real hyperspectral image data sets are employed to validate the superiority of the proposed method, and the experiment results show that our method outperforms several state-of-the-art methods.

In the future, we are committed to exploring better low-rank regularization to improve unmixing results more effectively. In terms of time complexity reduction, we consider employing parallel computing [71], such as multi-core CPU and GPU parallel computing, to improve computational efficiency.

**Author Contributions:** Y.Z. and F.W. wrote the manuscript; L.S. supervised and designed the experiments; F.W. performed the experiments and analyzed the results; L.S. analyzed the data; H.J.S. revised the paper.

**Funding:** This research was funded by the National Natural Science Foundation of China, grant number 61971233, 61672291, 61972206, 61672293, U1831127, in part by the PAPD (a project funded by the priority academic program development of Jiangsu Higher Education Institutions) fund, and in part by the 15th Six Talent Peaks Project in Jiangsu Province, grant number RJFW-015. The APC was funded by the Natural Science Foundation of the Jiangsu Higher Education Institutions of China, grant number 19KJB510040.

**Acknowledgments:** The authors would like to thank the editors and three anonymous reviewers for their constructive comments on our manuscript, which makes our work much better.

**Conflicts of Interest:** The authors declare no conflict of interest.

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
