# Peer review of "Sparse Unmixing for Hyperspectral Image with Nonlocal Low-Rank Prior"

_remotesensing, doi:10.3390/rs11242897_

Round 1

Reviewer 1 Report

Line 48-50: Algorithms mentioned are Endmember extraction algorithms, not unmixing.

Line 173: ASC constraint is indeed not the best one. However, why not applying sum to or less than 1 constraint?

Line 336: ANC instead of ASC?  

Figures 4, 9 and 11: Will it be possible to have the figures displayed in the same page?

Line 351: DS1 instead of DC1?

Line 361: Assess instead of assessment

Eq. 32: the numerator should have X^

Figures 6, 7, 8, 9: color bar varies from one map to another. please have it unified from 0 to 1. it may change the displays.

Line 410: Is there a reference to the mineral map produced by tricorder 3.3?

Section 4.2: Experiments with Real Data:

I don't think the analysis was enough, Why selecting only three classes comparison (Figure 11)? Besides, the authors could have chosen a different data set that is easier to analyze and have better reference data to compare with (Urban data set, Pavia University). that will give the chance to perform quantitative analysis too.

Table 4 could have been produced for the real data set (a fairly larger image than the simulated data). it will be interesting to see the results. 

Author Response

see the attahced word file

Author Response

see the attached word file

Reviewer 3 Report

Dear authors,

   This paper involves a new method for sparse unmixing for hyper spectral images with nonlocal low-rank prior. The research presented is good. My main concern is the presentation of the paper. There are lots of phrases that are grammatically incorrect or sound awkward. I suggest the authors go through the paper multiple times (consult a native speaker if necessary). The paper is definitely in need of proofreading.

    For example, the first sentence of the Introduction is grammatically incorrect. Other examples include (but NOT all of the cases):

1) Line 51, page 2, "few prior knowledges"

2) Line 72, page 2, "to constructs end member"

3) Line 107, page 3, "and then denoising the image"

4) Line 110, page 3, "imposed super pixel segmentation" 

5) Line 118, page 3, "unmoving algorithms shows"

6) Line 158, page 4, "scene, Let"

7) Line 185, page 5, "X shows the characteristics of sparse"

8) Line 182, page 5, "in practical application"

9) Line 192, page 6, "which describe" (should be describes)

10) Line 211, page 6,  "as a natural image"

11) Line 212, page 6, "The adjacent pixels are continuous transformation to some extent"

12) Line 241, page 7, "the any two spectral"

13) Line 297, page 9, "the optimization problem about V2"

14) Line 306 and 314, page 10, "the optimization problem about V4"..."about V6"

15) Line 304, page 10, "can be able to calculated"

16) Line 334, page 11, "by randomly selected"

17) Line 342, page 11, "ture fractional abundances"

18) Line 344, page 12, "ture fractional abundances"

19) Line 427, page 17, "We did several experiments"- awkward- say "we performed"

20) Line 432, page 18, "patch size for all data set, also 5 for the patch number" rewrite...

21) rewrite the last paragraph of the conclusion. grammatically incorrect.

Other suggestions include

1) not splitting Algorithm 1 between two pages

2) including a separate section about time complexity right before section 4

3) not splitting Figures 4, 9 and 11 between two pages 

4) It seems that the method is the slowest. Include a justification for the use of the proposed algorithm right after the timing is described.

5) The last part of the Introduction talks about the advantages of the proposed method. This can be made even more explicit by bullet points or even a separate section. You need to convince the reader quickly and clearly as to why they would want to use this method. 

Author Response

see the attached word file

Round 2

Reviewer 1 Report

I'm satisfied with the changes and I believe the paper is ready to be published. 

Author Response

Thank you for positive comments.

Reviewer 2 Report

Please see an attached file.

Author Response

please see the file attached

Reviewer 3 Report

I am satisfied with the revisions and believe that the paper should now be accepted.

Author Response

Thank you for positive comments.